# DOS: Diverse Outlier Sampling for Out-of-distribution Detection

**Wenyu Jiang**[1, 2]*, **Hao Cheng**[2], **Mingcai Chen**[2], **Chongjun Wang**[2], **Hongxin Wei**[1]†

[1]Department of Statistics and Data Science, Southern University of Science and Technology
[2]State Key Laboratory for Novel Software Technology, Nanjing University

## Abstract

Modern neural networks are known to give overconfident predictions for out-of-distribution inputs when deployed in the open world. It is common practice to leverage a surrogate outlier dataset to regularize the model during training, and recent studies emphasize the role of uncertainty in designing the sampling strategy for outlier datasets. However, the OOD samples selected solely based on predictive uncertainty can be biased towards certain types, which may fail to capture the full outlier distribution. In this work, we empirically show that diversity is critical in sampling outliers for OOD detection performance. Motivated by the observation, we propose a straightforward and novel sampling strategy named DOS (Diverse Outlier Sampling) to select diverse and informative outliers. Specifically, we cluster the normalized features at each iteration, and the most informative outlier from each cluster is selected for model training with absent category loss. With DOS, the sampled outliers efficiently shape a globally compact decision boundary between ID and OOD data. Extensive experiments demonstrate the superiority of DOS, reducing the average FPR95 by up to 25.79% on CIFAR-100 with TI-300K.

## 1 Introduction

Modern machine learning systems deployed in the open world often fail silently when encountering out-of-distribution (OOD) inputs (Nguyen et al., 2015) – an unknown distribution different from in-distribution (ID) training data, and thereby should not be predicted with high confidence. A reliable classifier should not only accurately classify known ID samples, but also identify as "unknown" any OOD input. This emphasizes the importance of OOD detection, which determines whether an input is ID or OOD and allows the model to raise an alert for safe handling.

To alleviate this issue, it is popular to assume access to a large auxiliary OOD dataset during training. A series of methods are proposed to regularize the model to produce lower confidence (Hendrycks et al., 2019b; Mohseni et al., 2020) or higher energy (Liu et al., 2020) on the randomly selected data from the auxiliary dataset. Despite the superior performance over those methods without auxiliary OOD training data, the random sampling strategy yields a large portion of uninformative outliers that do not benefit the differentiation of ID and OOD data (Chen et al., 2021), as shown in Figure 1a & 1b. To efficiently utilize the auxiliary OOD dataset, recent works (Chen et al., 2021; Li & Vasconcelos, 2020) design greedy sampling strategies that select hard negative examples, i.e., outliers with the lowest predictive uncertainty. Their intuition is that incorporating hard negative examples may result in a more stringent decision boundary, thereby improving the detection of OOD instances. However, the OOD samples selected solely based on uncertainty can be biased towards certain classes or domains, which may fail to capture the full distribution of the auxiliary OOD dataset. As shown in Figure 1c, the concentration of sampled outliers in specific regions will result in suboptimal performance of OOD detection (see Section 2.2 for more details). This motivates us to explore the importance of diversity in designing sampling strategies.

In this work, we empirically show that diversity is critical in designing sampling strategies, by the observation that outlier subset comprising data from more clusters results in better OOD detection. It

---

*Work done while working at SUSTech as a visiting scholar.
†Corresponding author (weihx@sustech.edu.cn)

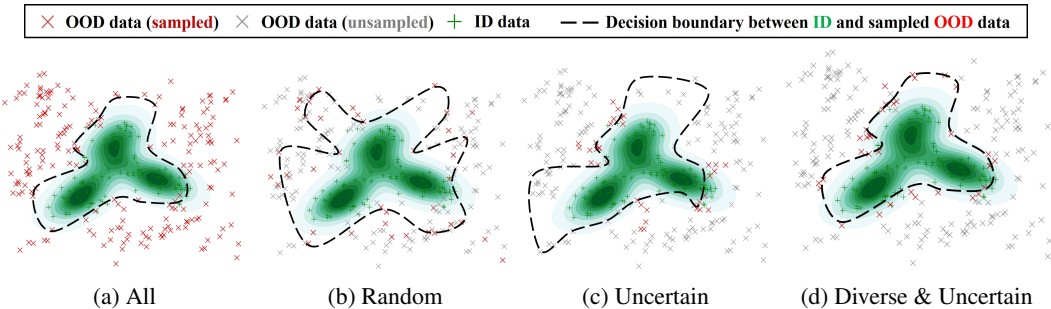

Figure 1: A toy example in 2D space for illustration of different sampling strategies. The ID data consists of three class-conditional Gaussian distributions, and the OOD training samples are simulated with plenty of small-scale class-conditional Gaussian distributions away from ID data. (a): All outliers sampled: a global compact boundary but intractable. (b): Random outliers sampled: efficient, with a loose boundary. (c): Uncertain outliers sampled: efficient, with a locally compact boundary (See Subsection 2.2 for more empirical results). (d): Diverse and uncertain outliers sampled: efficient, with a globally compact boundary.

is noteworthy that the diverse outlier pool without considering the cost of development (Hendrycks et al., 2019b) might not directly transfer to the outlier subset, due to the deficient sampling strategy. Therefore, the sampling strategy should improve the diversity of selected hard negative samples, for a globally compact decision boundary as shown in Figure 1d. Specifically, we propose a straight-forward and novel sampling strategy named DOS (**D**iverse **O**utlier **S**ampling), which first clusters the candidate OOD samples, and then selects the most informative outlier from each cluster, without dependency on external label information or pre-trained model. For efficient and diverse clustering, we utilize the normalized latent representation in each iteration with the K-means algorithm. Trained with absent category loss, the most informative outlier can be selected from each cluster based on the absent category probability. In this way, a diverse and informative outlier subset efficiently unlocks the potential of an auxiliary OOD training dataset.

To verify the efficacy of our sampling strategy, we conduct extensive experiments on common and large-scale OOD detection benchmarks, including CIFAR-100 (Krizhevsky & Hinton, 2009) and ImageNet-1K (Deng et al., 2009) datasets. Empirical results show that our method establishes *state-of-the-art* performance over existing methods for OOD detection. For example, using CIFAR-100 dataset as ID and a much smaller TI-300K (Hendrycks et al., 2019b) as an auxiliary OOD training dataset, our approach reduces the FPR95 averaged over various OOD test datasets from 50.15% to 24.36% – a **25.79%** improvement over the NTOM method, which adopts greedy sampling strategy (Chen et al., 2021). Moreover, we show that DOS keeps consistent superiority over other sampling strategies across different auxiliary outlier datasets and regularization terms, such as energy loss (Liu et al., 2020). In addition, our analysis indicates that DOS works well with varying scales of the auxiliary OOD dataset and thus can be easily adopted in practice.

In Section 5, we perform in-depth analyses that lead to an improved understanding of our method. In particular, we contrast with alternative features in clustering and demonstrate the advantages of feature normalization in DOS. While the clustering step introduces extra computational overhead, we find that DOS can benefit from faster convergence, leading to efficient training. Additionally, we demonstrate the value of OE-based methods in the era of large models by boosting the OOD detection performance of CLIP. We hope that our insights inspire future research to further explore sampling strategies for OOD detection.

## 2 PRELIMINARIES

### 2.1 BACKGROUND

**Setup.** In this paper, we consider the setting of supervised multi-class image classification. Let $\mathcal{X} = \mathbb{R}^d$ denote the input space and $\mathcal{Y} = \{1, ..., K\}$ denote the corresponding label space. The training dataset $\mathcal{D}_{\text{in}}^{\text{train}} = \{(\boldsymbol{x}_i, y_i)\}_{i=1}^N$ is drawn *i.i.d* from the joint data distribution $\mathbb{P}_{\mathcal{X} \times \mathcal{Y}}$. We use

$\mathbb{P}_{\mathcal{X}}^{\text{in}}$ to denote the marginal probability distribution on $\mathcal{X}$, which represents the in-distribution (ID). Given the training dataset, we learn a classifier $f_\theta : \mathcal{X} \mapsto \mathbb{R}^{|\mathcal{Y}|}$ with learnable parameter $\theta \in \mathbb{R}^p$, to correctly predict label $y$ of input $\boldsymbol{x}$. Let $\boldsymbol{z}$ denote the intermediate feature of $\boldsymbol{x}$ from $f_\theta$.

**Problem statement.** During the deployment stage, the classifier in the wild can encounter inputs from an unknown distribution, whose label set has no intersection with $\mathcal{Y}$. We term the unknown distribution out-of-distribution (OOD), denoted by $\mathbb{P}_{\mathcal{X}}^{\text{out}}$ over $\mathcal{X}$. The OOD detection task can be formulated as a binary-classification problem: determining whether an input $\boldsymbol{x}$ is from $\mathbb{P}_{\mathcal{X}}^{\text{in}}$ or not ($\mathbb{P}_{\mathcal{X}}^{\text{out}}$). OOD detection can be performed by a level-set estimation:

$$g(x) = \begin{cases} \text{in}, & \text{if } S(\boldsymbol{x}) \geq \tau \\ \text{out}, & \text{if } S(\boldsymbol{x}) < \tau \end{cases}$$

where $S(\boldsymbol{x})$ denotes a scoring function and $\tau$ is a threshold, which is commonly chosen so that a high fraction (e.g., 95%) of ID data is correctly distinguished. By convention, samples with higher scores are classified as ID and vice versa.

**Auxiliary OOD training dataset.** To detect OOD data during testing, it is popular to assume access to an auxiliary unlabeled OOD training dataset $\mathcal{D}_{\text{out}}^{\text{aux}} = \{\boldsymbol{x}\}_{i=1}^M$ from $\mathbb{P}_{\mathcal{X}}^{\text{out}}$ at training stage ($M \gg N$). In particular, the auxiliary dataset $\mathcal{D}_{\text{out}}^{\text{aux}}$ is typically selected independently of the specific test-time OOD datasets denoted by $\mathcal{D}_{\text{out}}^{\text{test}}$. For terminology clarity, we refer to training-time OOD data as outlier and exclusively use OOD data to refer to test-time unknown inputs.

To leverage the outliers from auxiliary datasets $\mathcal{D}_{\text{out}}^{\text{aux}}$, previous works (Hendrycks et al., 2019b; Mohseni et al., 2020; Liu et al., 2020) propose to regularize the classifier to produce lower scores on the randomly selected outliers. Formally, the objective can be formulated as follows:

$$\mathcal{L} = \mathbb{E}_{(\boldsymbol{x},y) \sim \mathcal{D}_{\text{in}}^{\text{train}}} \left[ \mathcal{L}(f(\boldsymbol{x}), y) + \lambda \mathbb{E}_{\boldsymbol{x} \sim \mathcal{D}_{\text{out}}^{\text{aux}}} \left[ \mathcal{L}_{\text{OE}}(f(\boldsymbol{x}), y) \right] \right] \tag{1}$$

However, the random sampling strategy yields a large portion of uninformative outliers that do not benefit the OOD detection (Chen et al., 2021). Recent works (Li & Vasconcelos, 2020; Chen et al., 2021) designed greedy strategies to sample outliers with the lowest predictive uncertainty, thus resulting in a more stringent decision boundary. Despite the superior performance of greedy strategies over those methods without auxiliary OOD training data, the OOD samples selected solely based on uncertainty can be biased towards certain classes or domains, which may fail to capture the full distribution of the auxiliary OOD dataset. In the following section, we empirically show the bias of greedy sampling and reveal the importance of diversity in designing sampling strategies.

## 2.2 Motivation

To demonstrate the inherent bias of the greedy sampling strategy, we divide the auxiliary dataset into multiple groups based on semantic information. In particular, we adopt the K-means to group similar outliers with their intermediate features, extracted by a pre-trained model. For the greedy sampling, we select outliers with the highest predictive confidence following ATOM (Chen et al., 2021). For comparison, we provide two additional sampling strategies: *uniform sampling* that uniformly samples outliers from different groups and *biased sampling* that selects outliers from only a group. We construct three subsets with the same size using the three sampling strategies, respectively.

In this part, we perform standard training with DenseNet-101 (Huang et al., 2017), using CIFAR-100 (Krizhevsky & Hinton, 2009) as ID dataset and TI-300K (Hendrycks et al., 2019b) as outlier pool. For evaluation, we use the commonly used six OOD test datasets. To extract features for clustering, we use the pretrained WRN-40-2 (Zagoruyko & Komodakis, 2016) model (Hendrycks et al., 2019a). For the clustering, we set the number of clusters as 6. More experimental details can be found in Appendix B.

**The sampling bias of greedy strategy.** Figure 2a presents the clustering label distribution of outliers sampled by the greedy and uniform strategies in . The x-axis denotes the ID of different clusters. The results show that the greedy strategy leads to a biased sampling, which exhibits an imbalanced distribution of outliers over the six clusters. For example, the number of outliers from cluster C1 is nearly *twice* that of cluster C6. With imbalanced distribution, the biased outliers from greedy sampling may fail to capture the full distribution of the auxiliary OOD training dataset, which degrades the performance of OOD detection.

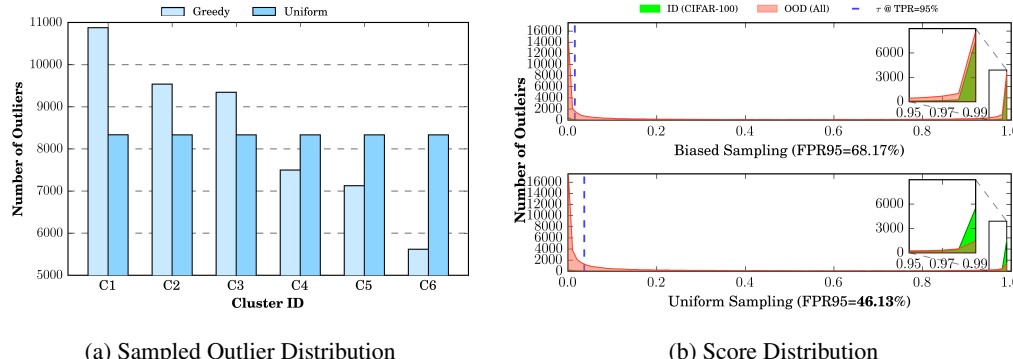

(a) Sampled Outlier Distribution                    (b) Score Distribution

Figure 2: Comparisons among different sampling strategies. (a): The outliers (TI-300K) distribution across six clustering centers with greedy and uniform strategies. (b): The score distribution for ID (CIFAR-100) and OOD (All) using biased and uniform strategies. Compared with uniform sampling, biased sampling produces more OOD examples with high scores that are close to ID.

**The importance of diversity in designing sampling strategies.** The formal definition of diversity can be found in Appendix E. To verify the effect of diversity in outlier sampling, we compare the OOD detection performance of models trained with the biased and uniform strategies, presented in Figure 2b. Here, we use the inverse of absent category probability as a scoring function. Recall that the biased strategy is an extreme example that selects outliers from only a cluster, the uniform strategy maximizes the diversity by uniformly selecting outliers from the six clusters. The results show that the uniform strategy with max diversity achieves a much lower FPR95 than the biased strategy, which demonstrates the critical role of diversity in sampling.

To understand how the diversity of outliers affects OOD detection, we compare the score distribution of the biased and uniform strategies in Figure 2b. We can observe that the biased sampling produces more OOD examples with high scores that are close to ID examples, making it challenging to differentiate the ID and OOD examples. This phenomenon aligns with the locally compact decision boundary shown in Figure 1c. In contrast, diverse outliers selected by the uniform strategy result in smooth score distribution, and thus better differentiation of ID and OOD data. In this way, we show that the diversity of outliers is a critical factor in designing sampling strategies.

## 3 METHOD: DIVERSE OUTLIER SAMPLING

From our previous analysis, we show that by training with outliers that are sufficiently diverse, the neural network can achieve consistent performance of OOD detection across the feature space. For a compact boundary between ID and OOD examples, the selected outliers should be also informative, i.e., close to ID examples (Chen et al., 2021). Inspired by the insights, our key idea in this work is to select the most informative outliers from multiple distinct regions. In this way, the selected outlier could contain sufficient information for differentiating between ID and OOD examples while maintaining the advantage of diversity.

To obtain distinct regions in the feature space of outliers, a natural solution is to utilize the semantic labels of the auxiliary dataset. However, it is prohibitively expensive to obtain annotations for such large-scale datasets, making it challenging to involve human knowledge in the process of division. To circumvent the issue, we present a novel sampling strategy termed Diverse Outlier Sampling (**DOS**), which partitions outliers into different clusters by measuring the distance to the prototype of each cluster. In the following, we proceed by introducing the details of our proposed algorithms.

**Clustering with normalized features** To maintain the diversity of selected outliers, we employ a non-parametric clustering algorithm - K-means, which partitions the outliers from the auxiliary dataset into $k$ clusters $\mathbf{C} = \{C_1, C_2, \ldots, C_k\}$ so as to minimize the within-cluster sum of squares. Formally, the objective of the vanilla K-means algorithm is to find: $\arg\min_{\mathbf{C}} \sum_{i=1}^{k} \sum_{\mathbf{x} \in C_i} \|\mathbf{z} - \boldsymbol{\mu}_i\|^2$, where $\boldsymbol{\mu}_i$ is the centroid of outliers from the cluster $C_i$.

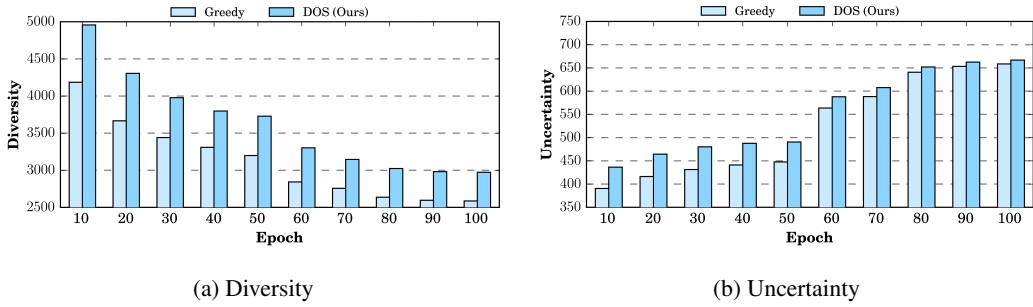

(a) Diversity                                    (b) Uncertainty

Figure 3: Comparison of the selected outliers between the greedy sampling and our proposed method in (a) diversity and (b) uncertainty. For the diversity, we adopt the label-independent clustering evaluation metric Calinski-Harabasz index (Caliński & Harabasz, 1974), which is the ratio of the sum of inter-cluster dispersion and of intra-cluster dispersion for all clusters. For the uncertainty, we use the softmax probability of the $(K{+}1)$-th class.

Nevertheless, adopting the vanilla K-means algorithm will introduce a bias towards features with larger scales, i.e., examples with confident predictions. In other words, those features with large scales may have a greater impact on the clustering process, which degrades the performance of outlier clustering. To address this issue, we propose to normalize the features before the clustering, thereby mitigating the negative effect of the feature scale. We provide an analysis in Section 5 to validate the effect of the normalization in clustering. In particular, the new objective of the normalized K-means algorithm is:

$$\arg \min_{\mathbf{C}} \sum_{i=1}^{k} \sum_{\mathbf{x} \in C_i} \left\| \frac{\mathbf{z}}{\|\mathbf{z}\|} - \boldsymbol{\mu}_i \right\|^2$$

Now, we can partition outliers from the auxiliary dataset into $k$ clusters with the normalized K-means algorithms. By uniformly sampling from these clusters, the diversity of the selected outlier can be easily bounded. In Appendix D, we provide an ablation study to show the effect of the number of clustering centers and the choice of clustering algorithm.

**Active sampling in each cluster**    Despite that using diverse outliers can promote a balanced sampling, the selected outliers might be too easy for the detection task, which cannot benefit the differentiation of ID and OOD data. Therefore, it is important to filter out those informative outliers from each cluster. Following the principle of greedy sampling, we select the hard negative examples that are close to the decision boundary. Practically, we use the inverse absent category probability as a scoring function and select the outlier with the highest score in each cluster. For cluster $C_i$, the selected outlier $\boldsymbol{x}_j$ is sampled by

$$\arg \max_{\mathbf{j}} \left[ 1.0 - p(K + 1 | \boldsymbol{x}_j) \right].$$

With the diverse and informative outliers, the model could shape a globally compact decision boundary between ID and OOD data, enhancing the OOD detection performance.

**Mini-batch scheme**    Previous works (Chen et al., 2021; Ming et al., 2022b) normally sample the outliers from the candidate pool in the epoch level. However, the overwhelming pool heavily slows down the clustering process. For efficient sampling, we design a mini-batch scheme by splitting the full candidate pool into small groups sequentially. In each iteration, we select outliers by the proposed sampling strategy to regularize the model.

It is intuitive to utilize feature visualization for data diversity verification. However, the auxiliary OOD training data distribution is broad, and different sampling strategies may have minor differences, which is hard to perceive qualitatively. Therefore, we choose to quantify and compare the diversity and uncertainty differences. As shown in Figure 3a, the outliers selected from our sampling strategy indeed achieve much larger diversity than those of the greedy strategy. The results of

Table 1: OOD detection results on common benchmark. All values are percentages. ↑ indicates larger values are better, and ↓ indicates smaller values are better. **Bold** numbers are superior results.

| $D_{\text{out}}^{\text{train}}$ | Method | FPR95 ↓ / AUROC ↑ | | | | | | | ACC |
| | | $D_{\text{out}}^{\text{test}}$ | | | | | | | |
| | | SVHN | LSUN-C | DTD | Places | LSUN-R | iSUN | Average | |
| NA | MSP | 83.3 / 76.3 | 78.6 / 78.2 | 86.9 / 70.6 | 83.0 / 74.1 | 82.3 / 74.0 | 84.2 / 72.4 | 83.0 / 74.3 | **75.6** |
| | ODIN | 92.9 / 73.9 | 55.9 / 88.4 | 85.2 / 71.2 | 75.7 / 79.0 | 37.3 / 93.3 | 42.4 / 91.9 | 64.9 / 82.9 | 75.6 |
| | Maha | 47.4 / 88.4 | 77.3 / 71.1 | 30.4 / 91.6 | 94.2 / 57.9 | 23.7 / 95.3 | 23.3 / 95.2 | 49.4 / 83.3 | 75.6 |
| | Energy$_{\text{score}}$ | 85.7 / 80.8 | 54.4 / 89.5 | 87.5 / 69.3 | 76.8 / 78.2 | 63.1 / 87.7 | 67.1 / 86.0 | 72.4 / 81.9 | 75.6 |
| | ReAct | 83.8 / 81.4 | 25.6 / 94.9 | 77.8 / 79.0 | 82.7 / 74.0 | 60.1 / 87.9 | 65.3 / 86.6 | 62.3 / 84.5 | 75.6 |
| | DICE | 54.7 / 88.8 | 0.9 / 99.7 | 65.0 / 76.4 | 79.6 / 77.3 | 49.4 / 91.0 | 48.7 / 90.1 | 49.7 / 87.2 | 75.6 |
| TI300K | OE | 42.6 / 91.5 | 34.1 / 93.6 | 57.0 / 87.3 | 54.7 / 87.3 | 45.7 / 90.8 | 48.6 / 90.0 | 47.1 / 90.1 | 74.4 |
| | Energy$_{\text{loss}}$ | 20.6 / 96.4 | 26.0 / 95.2 | 53.4 / 88.4 | **53.2 / 88.9** | 34.5 / 93.0 | 37.7 / 92.3 | 37.7 / 92.3 | 68.7 |
| | NTOM | 28.4 / 95.2 | 36.5 / 93.7 | 52.5 / 88.7 | 60.9 / 87.9 | 59.9 / 86.2 | 62.7 / 84.7 | 50.2 / 89.4 | 74.6 |
| | Share | 27.8 / 95.3 | 28.5 / 94.9 | 47.7 / 89.3 | 54.9 / 88.7 | 38.1 / 92.5 | 42.2 / 91.3 | 39.9 / 92.0 | 75.1 |
| | POEM | 63.4 / 88.1 | 38.5 / 92.7 | 57.2 / 87.9 | 59.5 / 80.5 | 57.2 / 87.6 | 58.7 / 86.1 | 55.7 / 87.1 | 63.4 |
| | **DOS (Ours)** | **13.1 / 97.4** | **20.0 / 96.4** | **34.9 / 92.3** | 59.6 / 88.6 | **8.2 / 98.4** | **10.3 / 97.8** | **24.4 / 95.2** | 75.5 |
| INRC | SOFL | 21.5 / 96.2 | 17.4 / 96.7 | 57.0 / 87.4 | 60.5 / 87.6 | 50.3 / 90.3 | 53.5 / 89.3 | 43.4 / 91.2 | 72.6 |
| | OE | 19.7 / 95.2 | 0.6 / 99.7 | 8.8 / 97.1 | 30.9 / 91.6 | 0.0 / 100.00 | 0.0 / 99.9 | 10.0 / 97.3 | 73.7 |
| | ACET | 55.9 / 90.4 | 14.6 / 97.4 | 62.0 / 86.3 | 56.3 / 86.8 | 56.4 / 88.2 | 60.5 / 86.8 | 50.9 / 89.3 | 72.7 |
| | CCU | 50.8 / 91.6 | 12.0 / 97.8 | 60.8 / 86.3 | 55.2 / 87.2 | 38.4 / 91.8 | 41.0 / 90.9 | 43.0 / 91.0 | 74.6 |
| | ROWL | 98.9 / 50.3 | 88.7 / 55.4 | 97.0 / 51.2 | 98.9 / 50.3 | 88.3 / 55.6 | 90.4 / 54.5 | 93.4 / 53.0 | 72.5 |
| | Energy$_{\text{loss}}$ | 34.0 / 94.5 | 0.1 / 99.9 | 3.6 / 98.8 | 18.6 / 96.1 | 0.0 / 100.0 | 0.0 / 100.0 | 9.4 / 98.2 | 72.7 |
| | NTOM | 12.8 / 97.8 | 0.3 / 99.9 | 5.6 / 98.7 | 29.1 / 94.1 | 0.0 / 100.0 | 0.1 / 100.0 | 8.0 / 98.4 | 74.8 |
| | Share | 14.5 / 97.3 | 0.5 / 99.7 | 6.5 / 98.3 | 23.9 / 95.1 | 0.0 / 100.0 | 0.1 / 100.0 | 7.6 / 98.4 | **74.9** |
| | POEM | 18.6 / 96.7 | 0.1 / 99.9 | 3.3 / 98.9 | 19.5 / 95.7 | 0.0 / 100.0 | 0.0 / 100.0 | 6.9 / 98.5 | 72.4 |
| | **DOS (Ours)** | **4.0 / 98.9** | **0.0 / 99.9** | **2.3 / 99.3** | **12.7 / 97.4** | **0.0 / 100.0** | **0.0 / 100.0** | **3.2 / 99.3** | 74.1 |

Figure 3b show that our strategy can obtain outliers with comparable OOD scores to those of the greedy strategy, which demonstrates the informativeness of the selected outliers.

**Training objective** In each iteration, we use the mixed training set comprising labeled ID data and unlabeled outliers for training the neural network. Concretely, the classifier is trained to optimize $\boldsymbol{\theta}$ by minimizing the following cross-entropy loss function:

$$\mathcal{L} = \mathbb{E}_{(\boldsymbol{x},\boldsymbol{y})\sim\mathcal{D}_{\text{in}}^{\text{train}}}[-\boldsymbol{y}\log p(\boldsymbol{y}|\boldsymbol{x})] + \mathbb{E}_{\boldsymbol{x}\sim\mathcal{D}_{\text{out}}^{\text{sam}}}[-\log p(K+1|\boldsymbol{x})] \quad (2)$$

The details of DOS are presented in Appendix C. Our sampling strategy is a general method, orthogonal to different regularization terms, and can be easily incorporated into existing loss functions with auxiliary OOD datasets, e.g., energy loss (Liu et al., 2020). We provide a generality analysis in Section 4.2 to show the effectiveness of our method with energy loss.

## 4 EXPERIMENTS

In this section, we validate the effectiveness of DOS on the common and large-scale benchmarks. Moreover, we perform ablation studies to show the generality of DOS and the effect of the auxiliary OOD training dataset scales. Code is available at: `https://github.com/lygjwy/DOS`.

### 4.1 SETUP

**Datasets.** We conduct experiments on CIFAR100 (Krizhevsky & Hinton, 2009) as common benchmark and ImageNet-10 (Ming et al., 2022a) as large-scale benchmark. For CIFAR100, a down-sampled version of ImageNet (ImageNet-RC) (Chen et al., 2021) is utilized as an auxiliary OOD training dataset. Additionally, we use the 300K random Tiny Images subset (TI-300K)[1] as an alternative OOD training dataset, due to the unavailability of the original 80 Million Tiny Images[2] in previous work (Hendrycks et al., 2019b). The methods are evaluated on six OOD test datasets: SVHN (Netzer et al., 2011), cropped/resized LSUN (LSUN-C/R) (Yu et al., 2015), Textures (Cimpoi et al., 2014), Places365 (Zhou et al., 2017), and iSUN (Xu et al., 2015). More experimental setups can be found in Appendix B.

[1] `https://github.com/hendrycks/outlier-exposure`
[2] The original dataset contains offensive contents and is permanently downgraded.

Table 2: OOD detection results on large-scale benchmark. All values are percentages. ↑ indicates larger values are better, and ↓ indicates smaller values are better. **Bold** numbers are superior results.

| $D_{out}^{train}$ | Method | FPR95 ↓ / AUROC ↑ | | | | |
|---|---|---|---|---|---|---|
| | | $D_{out}^{test}$ | | | | |
| | | iNaturalist | SUN | Places | Texture | Average |
| NA | MSP | 54.99 / 87.74 | 70.83 / 80.86 | 73.99 / 79.76 | 68.00 / 79.61 | 66.95 / 81.99 |
| | ODIN | 47.66 / 89.66 | 60.15 / 84.59 | 67.89 / 81.78 | 50.23 / 85.62 | 56.48 / 85.41 |
| | Maha | 97.00 / 52.65 | 98.50 / 42.41 | 98.40 / 41.79 | 55.80 / 85.01 | 87.43 / 55.47 |
| | $Energy_{score}$ | 55.72 / 89.95 | 59.26 / 85.89 | 64.92 / 82.86 | 53.72 / 85.99 | 58.41 / 86.17 |
| | ReAct | **20.38** / 96.22 | 24.20 / 94.20 | 33.85 / 91.58 | 47.30 / 89.80 | 31.43 / 92.95 |
| | DICE | 25.63 / 94.49 | 35.15 / 90.83 | 46.49 / 87.48 | 31.72 / 90.30 | 34.75 / 90.77 |
| IN-990 | OE | 21.10 / **97.08** | 28.72 / 96.19 | 30.70 / 95.95 | 14.59 / 97.67 | 23.78 / 96.72 |
| | $Energy_{loss}$ | 35.77 / 95.44 | 40.05 / 94.43 | 42.29 / 94.04 | 27.96 / 96.02 | 36.52 / 94.98 |
| | NTOM | 61.75 / 94.16 | 45.41 / 94.98 | 43.94 / 94.93 | 59.10 / 94.43 | 52.55 / 94.62 |
| | Share | 29.05 / 95.90 | 30.52 / 95.45 | 30.38 / 95.26 | 23.97 / 96.40 | 28.48 / 95.75 |
| | DOS (Ours) | 22.78 / 96.80 | **24.0** / **96.29** | **25.31** / **96.11** | **10.76** / **97.84** | **20.71** / **96.76** |

Table 3: Results with energy loss.

| Under-sampling | FPR95 ↓ | AUROC ↑ |
|---|---|---|
| Random | 37.72 | 92.32 |
| NTOM | 84.68 | 52.44 |
| POEM | 55.74 | 87.14 |
| **DOS (Ours)** | **29.43** | **94.05** |

Table 4: FPR95 with varying OOD size.

| Method | 25% | 50% | 75% | 100% |
|---|---|---|---|---|
| OE | 26.12 | 24.2 | 22.18 | 23.78 |
| Energy | 32.45 | 30.61 | 29.87 | 36.52 |
| NTOM | 66.21 | 60.62 | 55.99 | 52.55 |
| Share | 26.12 | 29.87 | 29.87 | 28.48 |
| **DOS (Ours)** | **21.97** | **21.56** | **21.03** | **20.71** |

**Training setting.** We use DenseNet-101 for the common benchmark and DenseNet-121 for the large-scale benchmark. The model is trained for 100 epochs using SGD with a momentum of 0.9, a weight decay of 0.0001, and a batch size of 64, for both ID and OOD training data. The initial learning rate is set as 0.1 and decays by a factor of 10 at 75 and 90 epochs. The above settings are the same for all methods trained with auxiliary outliers. By default, the size of sampled OOD training samples is the same as the ID training dataset, which is a common setting in prior work (Ming et al., 2022b). Without tuning, we keep the number of the clustering center the same as the batch size. All the experiments are conducted on NVIDIA V100 and all methods are implemented with default parameters using PyTorch.

**Compared methods.** According to dependency on auxiliary OOD training dataset, we divide the comparison methods into (1) Post-hoc methods: MSP (Hendrycks & Gimpel, 2017), ODIN (Liang et al., 2018), Maha (Lee et al., 2018b), $Energy_{score}$ (Liu et al., 2020), GradNorm (Huang et al., 2021), ReACT (Sun et al., 2021), and DICE (Sun & Li, 2022), and (2) Outlier exposure methods: SOFL (Mohseni et al., 2020), OE (Hendrycks et al., 2019b), ACET (Hein et al., 2019), CCU (Meinke & Hein, 2019), ROWL (Sehwag et al., 2019), $Energy_{loss}$ (Liu et al., 2020), NTOM (Chen et al., 2021), Share (Bitterwolf et al., 2022), and POEM (Ming et al., 2022b).

## 4.2 RESULTS

**DOS achieves superior performance on the common benchmark.** In Table 1, we present the OOD detection results of the CIFAR100-INRC benchmark. It is obvious that the outlier exposure methods normally perform better than those post-hoc methods. It is vital that our method outperforms existing competitive OE-based methods, establishing *state-of-the-art* performance. For example, DOS further reduces the average FPR95 by **3.7%**, compared to the most competitive energy-regularized method POEM, despite the overwhelming INRC dataset resulting in the saturated performance on several OOD test datasets. Trained with the same absent category loss, DOS shows superiority over random (Share) and greedy (NTOM) strategies, with a **4.4%** and **4.8%** improvement on the average FPR95, respectively. At the same time, we maintain comparable accuracy. The average FPR95 standard deviation of our methods over 3 runs of different random seeds is 1.36%, and 0.15% for the CIFAR100-TI300K and CIFAR100-INRC, respectively.

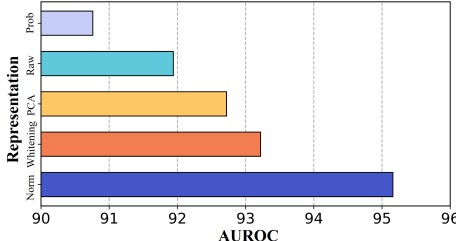

Figure 4: Results of different features.

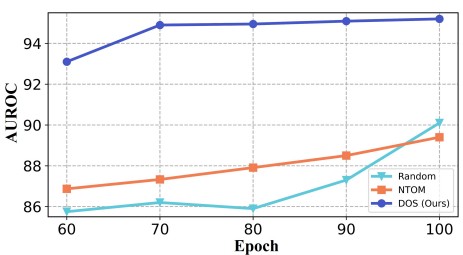

Figure 5: AUROC at varying epochs.

**DOS shows consistent superiority across different auxiliary OOD datasets.** To verify the performance of the sampling strategy across different auxiliary OOD training datasets, we treat TI-300K as alternative auxiliary outliers. Using the TI-300K dataset, the performance of compared methods largely deteriorated due to the small size of the auxiliary dataset. As shown in Table 1, the FPR95 of POEM degrades from 6.9% to 55.7%. However, our proposed method still shows consistent superiority over other methods. Specifically, the average FPR95 is reduced from 50.15% to 24.36% — a **25.79%** improvement over the NTOM method, which uses a greedy sampling strategy.

**DOS is effective on the large-scale benchmark.** We also verify the effectiveness of our method on the large-scale benchmark. Specifically, we use the ImageNet-10 as ID training dataset and ImageNet-990 as the auxiliary data. Table 2 presents the OOD detection results for each OOD test dataset and the average over the four datasets. We can observe that outlier exposure methods still demonstrate better differentiation between ID and OOD data than the post-hoc methods, and our method achieves the best performance, surpassing the competitive OE by **3.07%** in average FPR95. The average FPR95 standard deviation of DOS over 3 runs of different random seeds is **0.77%**.

**DOS shows generality and effectiveness with energy loss.** To validate the effectiveness of DOS with other regularization functions, we replace the original absent category loss with energy loss and detect OOD data with energy score (Liu et al., 2020). As shown in Table 3, we find that the proposed sampling strategy is consistently effective, establishing *state-of-the-art* performance over other sampling strategies. For example, on the CIFAR100-TI300K benchmark, using DOS sampling boosts the FPR95 of energy score from 37.72% to 29.43% — a **8.29%** of direct improvement.

**DOS is robust with varying scales of the auxiliary OOD dataset.** Here, we provide an empirical analysis of how the scale of auxiliary datasets affects the performance of DOS. We conduct experiments with the IN10-IN990 benchmark by comparing the performance with different percentages of the whole auxiliary OOD training dataset. As shown in Table 4, we can observe that DOS maintains superior OOD detection performance with all the outlier percentages. Even if we keep **25%** of the auxiliary OOD training dataset, the FPR95 of DOS only decreases by **1.26%**, which demonstrates the robustness of our method to the numbers of outliers.

## 5 DISCUSSION

**Feature processing for clustering.** In DOS, we normalize the features from the penultimate layer for clustering with K-means. While our method has demonstrated strong promise, a question arises: *Can a similar effect be achieved with alternative forms of features?* Here, we replace the normalized features with various representations, including softmax probabilities, raw latent features, and latent features with dimensionality reduction or whitening, in the CIFAR100-TI300K benchmark. In this ablation, we show that those alternatives do not work as well as our method.

Our results in Figure 4 show that employing normalized features achieves better performance than using softmax probabilities and raw features. While the post-processing methods, including PCA and whitening, can improve the performance over using raw features, feature normalization is still superior to those methods by a meaningful margin. Previous works demonstrated that OOD examples usually produce smaller feature norms than in-distribution data (Sun et al., 2022; Tack et al., 2020; Huang et al., 2021), which may disturb the clustering algorithm with Euclidean distance for

Table 5: FPR95 ↓ with the CLIP ViT model on large-scale benchmark. All values are percentages. ↓ indicates smaller values are better. **Bold** numbers are superior results.

| $D_{out}^{train}$ | Method | $D_{out}^{test}$ | | | | |
|---|---|---|---|---|---|---|
| | | iNaturalist | SUN | Places | Texture | Average |
| NA | MSP | 32.53 | 21.92 | 24.19 | 14.97 | 23.40 |
| | ODIN | 18.13 | 6.05 | 8.26 | 5.36 | 9.45 |
| | Maha | 22.31 | 8.22 | 10.02 | 4.52 | 11.27 |
| | $Energy_{score}$ | 26.20 | 8.83 | 11.96 | 7.98 | 13.74 |
| IN-990 | OE | 4.82 | 5.07 | 5.83 | 0.78 | 4.13 |
| | $Energy_{loss}$ | 19.60 | 7.81 | 9.35 | 3.42 | 10.04 |
| | NTOM | 9.33 | 7.45 | 9.28 | 3.69 | 7.44 |
| | Share | 5.85 | 7.05 | 8.18 | 1.56 | 5.66 |
| | **DOS (Ours)** | **4.00** | **4.08** | **4.82** | **0.75** | **3.41** |

distinct clustering. With normalization, the norm gap between ID and OOD examples can be diminished, leading to better performance in clustering. Empirically, we verify that normalization plays a key role in the clustering step of diverse outlier sampling.

**Fast convergence for efficiency.** Compared with random sampling, a potential limitation of DOS is the additional computational overhead from the clustering operation. Specifically, for the CIFAR100-TI300K benchmark, the clustering (0.0461s) takes a 12% fraction of the total training time (0.384s) for each iteration. On the other hand, we find the extra training cost can be covered by an advantage of our proposed method – fast convergence.

In Figure 5, we present the OOD detection performance of snapshot models at different training epochs. The results show that our method has established ideal performance at an early stage, while the performances of random sampling and NTOM are still increasing at the 100th epoch, with a significant gap to our method. Training with DOS, the model requires much fewer training epochs to achieve optimal performance, which demonstrates the superiority of our method in efficiency.

**Adaptation to pre-trained large models.** In recent works, large models have shown strong robustness to distribution shift by pretraining on broad data (Radford et al., 2021; Ming et al., 2022a). Here, we provide an empirical analysis to show *whether OE-based methods can help large models in OOD detection.* To this end, we conduct experiments by fine-tuning the pre-trained CLIP ViT (ViT-B/16) (Dosovitskiy et al., 2020) for OOD detection on the IN10-IN990 benchmark. For post-hoc methods, we only fine-tune the model on the in-distribution data, while we use both the ID data and the auxiliary dataset for OE-based methods. The model is fine-tuned for 10 epochs using SGD with a momentum of 0.9, a learning rate of 0.001, and a weight decay of 0.00001.

We present the results of the pre-trained CLIP model in Table 5. Indeed, pretrained large models can achieve stronger performance than models trained from scratch (shown in Table 2) in OOD detection. Still, fine-tuning with outliers can significantly improve the OOD detection performance of CLIP, where DOS achieves the best performance. Overall, the results demonstrate the value of OE-based methods in the era of large models, as an effective way to utilize collected outliers.

## 6 CONCLUSION

In this paper, we propose Diverse Outlier Sampling (DOS), a straightforward and novel sampling strategy. Based on the normalized feature clustering, we select the most informative outlier from each cluster, thereby resulting in a globally compact decision boundary between ID and OOD data. We conduct extensive experiments on common and large-scale OOD detection benchmarks, and the results show that our method establishes *state-of-the-art* performance for OOD detection with a limited auxiliary dataset. This method can be easily adopted in practical settings. We hope that our insights inspire future research to further explore sampling strategy design for OOD detection.

ACKNOWLEDGMENTS

This research is supported by the Shenzhen Fundamental Research Program (Grant No. JCYJ20230807091809020). Chongjun Wang is supported by the National Natural Science Foundation of China (Grant No. 62192783, 62376117). We gratefully acknowledge the support of the Center for Computational Science and Engineering at the Southern University of Science and Technology, and the Collaborative Innovation Center of Novel Software Technology and Industrialization at Nanjing University for our research.

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

## A  RELATED WORK

**OOD detection**   OOD detection is critical for the deployment of models in the open world. A popular line of research aims to design effective scoring functions for OOD detection, such as Open-Max score (Bendale & Boult, 2016), maximum softmax probability (Hendrycks & Gimpel, 2017), ODIN score (Liang et al., 2018), Mahalanobis distance-based score (Lee et al., 2018b), distance-based score with reconstruction error (Jiang et al., 2023), cosine similarity score (Hsu et al., 2020), Energy-based score (Liu et al., 2020; Wang et al., 2021; Morteza & Li, 2022), GradNorm score (Huang et al., 2021), and non-parametric KNN-based score (Sun et al., 2022). However, the model can be overconfident to the unknown inputs (Nguyen et al., 2015).

In order to investigate the fundamental cause of overconfidence, ReAct (Sun et al., 2021) proposes to rectify the extremely high activation. BATS (Zhu et al., 2022) further rectifies the feature into typical sets to achieve reliable uncertainty estimation. DICE (Sun & Li, 2022) exploits sparsification to eliminate the noisy neurons. LogitNorm (Wei et al., 2022) finds that the increasing norm of the logit leads to overconfident output, thereby a constant norm is enforced on the logit vector.

Another direction of work addresses the OOD detection problem by training-time regularization with auxiliary OOD training dataset (Mohseni et al., 2020; Hendrycks et al., 2019b; Hein et al., 2019; Meinke & Hein, 2019; Sehwag et al., 2019; Liu et al., 2020; Bitterwolf et al., 2022). The auxiliary OOD training dataset can be natural, synthetic (Lee et al., 2018a; Kong & Ramanan, 2021; Grcić et al., 2020), or mixed (Wang et al., 2023b). Models are encouraged to give predictions with uniform distribution (Hendrycks et al., 2019b) or higher energies (Liu et al., 2020) for outliers. To efficiently utilize the auxiliary OOD training dataset, recent works (Li & Vasconcelos, 2020; Chen et al., 2021) design greedy sampling strategies that select hard negative examples for stringent decision boundaries. POEM (Ming et al., 2022b) achieves a better exploration-exploitation trade-off by maintaining a posterior distribution over models. Recently, DAL (Wang et al., 2023a) augments the auxiliary OOD distribution with a Wasserstein Ball to approximate the unseen OOD distribution via augmentation. DivOE (Zhu et al., 2023) augments the informativeness of the auxiliary outliers via informative extrapolation, which generates new outliers closer to the decision boundary. In this work, we propose a straightforward and novel sampling strategy named DOS (Diverse Outlier Sampling), which first clusters the outliers, and then selects the most informative outlier from each cluster, resulting in a globally compact decision boundary.

**Coreset selection**   A closely related topic to our sampling strategy is coreset selection, which aims to select a subset of the most informative ID training samples. Based on the assumption that data points close to each other in feature space tend to have similar properties, geometry-based methods (Chen et al., 2012; Sener & Savarese, 2017; Sinha et al., 2020; Agarwal et al., 2020) try to remove those data points providing redundant information then the left data points form a coreset, which is similar to our method. However, geometry-based methods aim to select **in-distribution data** for better generalization. In this work, we select **outlier examples** as auxiliary data to improve OOD detection, which is different from active learning. In other words, the problem setting of this work is different from those methods.

## B  EXPERIMENTAL DETAILS

To clarify the motivation experiment setting in Section 2.2, we follow prior work (Hendrycks et al., 2019a) to train the model. Concretely, downsampled ImageNet, which is the 1000-class ImageNet dataset resized to 32×32 resolution, is used for pre-training. The pretrained model can be found at https://github.com/hendrycks/pre-training. Six OOD test datasets contain: SVHN (Netzer et al., 2011), cropped/resized LSUN (LSUN-C/R) (Yu et al., 2015), Textures (Cimpoi et al., 2014), Places365 (Zhou et al., 2017), and iSUN (Xu et al., 2015).

It is worth noting that we adopt center clipping to remove the black border of LSUN-C for a fair and realistic comparison. For reproducible results, we follow a fixed Places list from (Chen et al., 2021).

**The large-scale benchmark.**   To build the large-scale benchmark, we split the ImageNet-1K (Deng et al., 2009) into IN-10 (Ming et al., 2022a) and IN-990. Specifically, the IN-10 is the ID training dataset, and IN-990 serves as the auxiliary OOD training data. The OOD test datasets com-

prise (subsets of) iNaturalist (Van Horn et al., 2018), SUN (Xiao et al., 2010), Places (Zhou et al., 2017), and Textures (Cimpoi et al., 2014). For main results, the size of sampled OOD data is the same as the ID training dataset, which is a common setting (Ming et al., 2022b).

**Evaluation metrics.** We evaluate the performance of OOD detection by measuring the following metrics: (1) the false positive rate (FPR95) of OOD examples when the true positive rate of ID examples is 95%; (2) the area under the receiver operating characteristic curve (AUROC).

## C ALGORITHM

---
**Algorithm 1:** DOS: Diverse Outlier Sampling

---
**Input:** ID training dataset: $\mathcal{D}_{\text{in}}^{\text{train}}$, auxiliary OOD training dataset: $\mathcal{D}_{\text{out}}^{\text{aux}}$;
**Output:** OOD detector: $g$, classifier: $f$;

1 **for** $t \leftarrow 1$ **to** $T$ **do**
2     **Sample** random outliers from $\mathcal{D}_{\text{out}}^{\text{aux}}$ to get a candidate set $\mathcal{D}_{\text{out}}^{\text{can}}$;
3     **for** $(\mathcal{B}_{\text{in}}^{\text{train}}, \mathcal{B}_{\text{out}}^{\text{can}}) \subset (\mathcal{D}_{\text{in}}^{\text{train}}, \mathcal{D}_{\text{out}}^{\text{can}})$ **do**
4        **Calculate** OOD score list $\mathbb{V}$ on $\mathcal{B}_{\text{out}}^{\text{can}}$ using current classifier $f_\theta$;
5        **Fetch** normalized OOD features $\mathbb{Z}$ from extractor module $e_\theta$;
6        **Cluster** the $\mathbb{Z}$ into $\|\mathcal{B}_{\text{in}}^{\text{train}}\|$ centers;
7        **Add** the outlier with largest OOD score from each cluster into $\mathcal{B}_{\text{out}}^{\text{train}}$;
8        **Train** $f_\theta$ using the training objective of Equation 2 with $\mathcal{B}_{\text{in}}^{\text{train}}$ and $\mathcal{B}_{\text{out}}^{\text{train}}$;
9     **end**
10 **end**

---

## D ABLATION STUDIES

Table 6: OOD detection results with varying clustering algorithms. FPR95 and AUROC values are percentages. ↑ indicates larger values are better, and ↓ indicates smaller values are better. **Bold** numbers are superior results.

| Clustering | FPR95 ↓ / AUROC ↑ | | | | | Time |
|---|---|---|---|---|---|---|
| | $\mathcal{D}_{\text{out}}^{\text{test}}$ | | | | Average | |
| | iNaturalist | SUN | Places | Texture | | |
| Agglomerative | 44.08 / 94.83 | 40.16 / 94.70 | 40.19 / 94.61 | 29.11 / 96.56 | 38.39 / 95.18 | **0.05s** |
| K-Means++ seeding | 54.25 / 94.78 | 49.58 / 94.86 | 49.70 / 94.71 | 39.22 / 96.20 | 48.19 / 95.13 | 2.0s |
| Spherical K-Means | 27.46 / 96.22 | 26.86 / 96.12 | 27.93 / 95.95 | 14.18 / 97.61 | 24.11 / 96.48 | 3.1s |
| moVMF-soft | - | - | - | - | - | 242.1s |
| moVMF-hard | - | - | - | - | - | 243.0s |
| **K-Means (Ours)** | **22.78 / 96.80** | **24.0 / 96.29** | **25.31 / 96.11** | **10.76 / 97.84** | **20.71 / 96.76** | 4.5s |

**Sensitivity analysis on clustering algorithm.** In this section, we justify the selection of K-Means by comparing the performance of various clustering algorithms, including Agglomerative (Murtagh & Legendre, 2014), K-Means++ seeding (Arthur & Vassilvitskii, 2007; Li et al., 2023), Spherical K-Means, moVMF soft, and moVMF hard (Banerjee et al., 2005). The experiments are conducted with the large benchmark (In10-IN990). We keep the same training setting as in the manuscript. In addition to the OOD detection performance, we also present clustering time in each iteration, which is also a key factor of choosing clustering algorithm. As shown in the Table 6, K-means (our choice) achieves the best performance among the compared algorithms. In addition, the two algorithms based on moVMF require too much time in each iteration, making it non-trivial to implement in this case. The resting algorithms cannot achieve comparable performance to ours, while they three obtains a little higher efficiency. Therefore, we choose the vanilla K-Means with superior performance and acceptable cost.

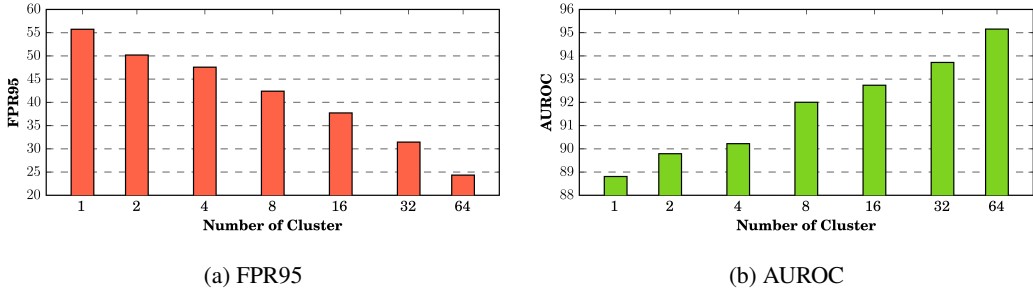

(a) FPR95                   (b) AUROC

Figure 6: OOD detection performance with a varying number of clusters.

**Effect of clustering centers number.** In this section, we emphasize the importance of diversity by adjusting the number of clustering centers. By default, the candidate OOD data pool is clustered into $K$ centers in each iteration, which is equal to the batch size of ID samples. As shown in Figure 6a & 6b, the OOD detection performance deteriorates as the number of clustering centers decreases. It is noted that the number of clustering centers is not a hyperparameter.

## E    DEFINITION OF DIVERSITY

Given a set of all OOD examples $\mathcal{D} = \{\mathbf{x}_i\}_{i=1}^{N}$, our goal is to select a subset of $K$ examples $\mathcal{S} \subseteq \mathcal{D}$ with $|\mathcal{S}| = K$. We define that the selected subset is diverse, if $\mathcal{S}$ satisfies the condition:

$$\delta(\mathcal{S}) = \frac{1}{K} \sum_{x_i \in \mathcal{S}} \min_{j \neq i} d\left(x_i, x_j\right) \geq C,$$

where $d$ denotes a distance function in the space and $C$ is a constant that controls the degree of diversity. A larger value of $C$ reflects a stronger diversity of the subset.

