# OpenReview forum: "DOS: Diverse Outlier Sampling for Out-of-Distribution Detection"
_ICLR.cc/2024/Conference — ICLR 2024 poster_

### Official Review · Reviewer_BvLD · 2023-10-29

**Soundness:** 3 good
**Presentation:** 3 good
**Contribution:** 2 fair
**Rating:** 6
**Confidence:** 3

**Summary:**

The paper proposes to use a diverse sampling to draw OOD samples from auxiliary datasets for training OOD models. The sampling consists of two steps. First, they apply K-means to cluster the auxiliary samples in a normalized feature space. Second, the samples close to the decision boundary of in-distribution are selected from each cluster. The clustering ensures the diversity of the selected samples. In training, the selected samples are considered OOD samples. They evaluate the proposed method on the common benchmark, i.e., CIFAR100, and show that the proposed diverse sampling improves over greedy sampling and other baselines.

**Strengths:**

1. The paper studies a well-motivated problem in OOD detection. Finding high-quality samples in the large auxiliary dataset improves the performance of the trained model and reduces the training cost.

2. The proposed diverse sampling is simple and effective.

3. The proposed method achieves impressive empirical results on the common benchmark.

4. The authors provide extensive ablation studies to demonstrate the robustness of the method.

**Weaknesses:**

1. Diverse sampling is a well-known method in active learning [1,2]. Diverse sampling leading to superior performance is not surprising. It indeed improves the performance of OOD models. Critically speaking, the novelty in terms of the method is limited. If the authors could provide some deeper theoretical analysis, e.g., how diverse sampling improves the generalization bound, it would make the paper more solid.

2. There are typos in equation 2. I understand that the authors use a cross-entropy loss. However, equation 2 is just the entropy. $p(y)$ or $y$ is missing inside the expectation.

[1] Ozan Sener and Silvio Savarese. Active learning for convolutional neural networks: A core-set approach. In
International Conference on Learning Representations, 2018.

[2] Jordan T Ash, Chicheng Zhang, Akshay Krishnamurthy, John Langford, and Alekh Agarwal. Deep batch active learning by diverse, uncertain gradient lower bounds. In International Conference on Learning Representations, 2020

**Questions:**

How important is the clustering step in diverse sampling? [1] proposed to use a K-means++ seeding algorithm based diverse sampling for semi-supervised anomaly detection. The K-means++ seeding algorithm doesn't require to predefine the clusters and the number of clusters. I think it can be also easily combined with the absent category probability. How would the proposed diverse sampling compare to the K-means++ seeding algorithm based diverse sampling?

[1] Li, Aodong, Chen Qiu, Marius Kloft, Padhraic Smyth, Stephan Mandt, and Maja Rudolph. "Deep anomaly detection under labeling budget constraints." In International Conference on Machine Learning, pp. 19882-19910. PMLR, 2023.

---

> ### Author Response · Authors · 2023-11-19
> **Response to Reviewer BvLD**
>
> We appreciate the reviewer for the insightful and detailed comments. Please find our response below:
> ## 1. Novelty and theoretical analysis [W1]
> Yes, the factor of diversity has been explored in the context of active learning, where the goal is to select **in-distribution data** for better generalization. In this work, we select **outlier examples** as auxiliary data to improve OOD detection, which is different from active learning. In other words, the problem setting of this work is different from those methods for active learning.
>
> To the best of our knowledge, this work is **the first to investigate the benefits of diversity in outlier sampling for OOD detection**, as recognized by Reviewer HJAj. In the literature, previous works proposed random selection [1,4,5,6] and greedy sampling [2,3] strategies to build the auxiliary OOD dataset. As shown in Section 5, our proposed method achieves substantial improvements over previous methods and enjoys fast convergence.
>
> This work also provides unique insights compared to the diverse sampling of active learning. In active learning, diverse sampling aims to find a collection of samples that can approximate the entire data distribution (ID). In this work, we propose the importance of diversity in alleviating the sampling bias of greedy sampling (See Subsection 2.2). As suggested by reviewer HJAj, we provide a formal definition of diversity (see response to reviewer HJAj), which is different from that of active learning.
>
> Lastly, we would like to clarify that outlier exposure is not directly linked to the generalization bound, which is usually analyzed in active learning. The goal of outlier exposure is to increase the differentiation between ID and OOD inputs, while we only have access to a finite OOD dataset from the infinite OOD feature space. As far as we know, there is no general theoretical framework in existing works to analyze outlier exposure. Instead, **we provide in-depth analysis to show the advantages of DOS from various perspectives**, as appreciated by all reviewers.
>
>
> ## 2. Comparison with K-means++ clustering algorithm [Q1]
> Thank you for the great suggestion. To investigate the importance of the clustering step in diverse sampling, we conduct sensitivity analysis to compare the performance of various clustering algorithms, including Agglomerative [7], K-Means++ seeding [8,9], Spherical K-Means, moVMF soft, and moVMF hard [10,11]. The experiments are conducted with a large benchmark (In10-IN990). We keep the same training setting as in the manuscript. In addition to the OOD detection performance, we also present clustering time in each iteration, which is also a key factor in choosing a clustering algorithm.
>
> As shown in the Table below, K-means (our choice) achieves the best performance among the compared algorithms. In addition, the two algorithms based on moVMF require too much time in each iteration, making it non-trivial to implement in this case. The resting algorithms cannot achieve comparable performance to ours, while the three obtain a little higher efficiency. We add the empirical results to Appendix D of the manuscript.
>
>
> |         Metric         | Agglomerative | K-Means++ seeding | Spherical K-Means | moVMF soft | moVMF hard | K-Means (Ours) |
> |:----------------------:|:-------------:|:-----------------:|:-----------------:|:----------:|:----------:|:--------------:|
> |   FPR95 $\downarrow$   |     38.39     |       48.19       |       24.11       |     -      |     -      |   **20.71**    |
> | Time Cost $\downarrow$ |   **0.05s**   |       2.0s        |       3.1s        |   242.1s   |   243.0s   |      4.5s      |
>
>
>
> ## 3. Typo in the euqation 2 [W2]
> Thank you for pointing it out. We have fixed it in the updated version.
>
> [1] Deep Anomaly Detection with Outlier Exposure. ICLR'19.
>
> [2] Background Data Resampling for Outlier-aware Classification. CVPR'20.
>
> [3] ATOM: Robustifying Out-of-distribution Detection Using Outlier Mining. ECML-PKDD'21.
>
> [4] Self-supervised Learning for Generalizable Out-of-distribution Detection. AAAI'20.
>
> [5] Energy-based Out-of-distribution Detection. NeurIPS'20.
>
> [6] Breaking down Out-of-distribution Detection: Many Methods based on OOD Training Data Estimate a Combination of the Same Core Quantities. ICML'22.
>
> [7] Ward’s Hierarchical Agglomerative Clustering Method: Which Algorithms Implement Ward’s Criterion? Journal of Classification 2014.
>
> [8] K-Means++: The Advantages of Careful Seeding. SODA'07.
>
> [9] Deep Anomaly Detection under Labeling Budget Constraints. ICML'23.
>
> [10] Clustering on the Unit Hypersphere using von Mises-Fisher Distributions. JMLR'05.
>
> [11] On PyTorch Implementation of Density Estimators for von Mises-Fisher and Its Mixture. arXiv'21.

---

> ### Author Response · Authors · 2023-11-22
> **Kind reminder to Reviewer BvLD**
>
> Dear Reviewer BvLD,
>
> Sorry to disturb you. We sincerely appreciate your valuable comments. We understand that you may be too busy to check our rebuttal. To avoid ambiguity, we would like to further provide a comparison of rigorous definitions of diversity, to show the novelty.
>
> Regarding the novelty, we would like to clarify the difference between our method and diverse sampling in active learning from the following perspectives:
> - **Problem Setting**: We select outlier examples as auxiliary data to improve OOD detection, while active learning aims to select in-distribution data for better generalization.
> - **Definition of Diversity**: Given a set of all examples $\mathcal{D} = { \lbrace \mathbf{x}\_i\rbrace }^{N}\_{i=1}$, we define that the selected subset $\mathcal{S}$ is diverse, if $\mathcal{S}$ satisfies the condition: $$\delta(\mathcal{S}) = \frac{1}{K}\sum\_{x\_{i} \in \mathcal{S}} \min \_{j \neq i} d\left(x\_{i}, x\_{j}\right) \geq C,$$ where $d$ denotes a distance function in the space, $C$ is a constant that controls the degree of diversity (see response to reviewer HJAj for more details). Our goal is to increase the lower bound $C$ while keeping the informativeness.
> Differently, the diversity sampling in active learning usually aims to minimize the following objective to construct $\mathcal{S}$: $$f(\mathcal{S})=\sum_{x_i \in \mathcal{D}} \min _{x_j \in \mathcal{S}} d\left(x_i, x_j\right).$$ Their goal is to minimize the $f(\mathcal{S})$ so that the subset $\mathcal{S}$ is sufficiently representative to replace $\mathcal{D}$.
> - **Insight**: In active learning, diverse sampling aims to find a collection of samples that can approximate the entire data distribution (ID). In this work, we propose the importance of diversity in alleviating the sampling bias of greedy sampling (See Subsection 2.2).
>
> In addition, we have fixed the typo in equation 2 and conducted additional sensitivity analysis to compare the performance of various clustering algorithms, as you suggested. We show that K-means (our choice) achieves the best performance among the compared algorithms. The detailed results can be found in Appendix D.
>
> We sincerely hope that the above answers can address your concerns. We look forward to your response and are willing to answer any questions.

---

### Official Review · Reviewer_HJAj · 2023-10-30

**Soundness:** 4 excellent
**Presentation:** 3 good
**Contribution:** 3 good
**Rating:** 8
**Confidence:** 4

**Summary:**

This paper focuses on outlier sampling for out-of-distribution detection (OOD detection) tasks. To be specific, this paper follows the setting of outlier exposure that utilizes a surrogate outlier dataset to regularize the model during training, trying to make the model better recognize those OOD inputs. This work points out that previous outlier sampling methods are solely based on predictive uncertainty which may fail to capture the full outlier distribution. Motivated by the empirical evidence which shows the criticality of diversity, this work proposes Diverse Outlier Sampling (DOS) to select diverse and informative outliers via clustering the normalized features at each iteration. The proposed method achieves an efficient way to shape the decision boundary between ID and OOD data. Experiments from different perspective are conducted to demonstrate the effectiveness of DOS.

**Strengths:**

1. This paper focuses on an important and practical question on outlier exposure, i.e., the auxiliary outliers may fail to capture the full outlier distribution.
2. This paper proposes a new method, namely DOS, which clusters the normalized features at each iteration and samples the informative outlier from each cluster to realize the diversified outlier selection. The technical design for clustering with normalized features is noval to the knowledge of the reviewer and shows promising empirical achievement towards the target.
3. Comprehensive experiments compared with both post-hoc OOD detection scores and also several representative sampling methods are conducted to demonstrate the effectiveness of the proposed DOS.
4. The overall presentation is clear and the method is easy to understand.

**Weaknesses:**

Overall, this work presents a concise and effective way to conduct diverse sampling in outlier exposure. Here are the major concerns for the current version of this paper, and hope it can help to improve the paper better.
1. Although the overall presentation is clear, some critical definitions and claims are questionable and lack of convincing support.
2. Technically, the proposed method (DOS) is based on an empirical demonstration of "diversity" with the OOD detection performance. However, the detailed definition of diversity is under-defined and the underlying mechanism of the clustering-based sampling scheme is not clearly explained.
3. Since the previous greedy strategy will continually sample those outliers that are easy to be recognized as ID data, why do we need to utilize the diverse sampling method if the newly proposed method will sample some outliers that are already recognized as OOD data by the model?
4. The experimental part can include more results conducted in other ID datasets, as well as the large benchmark dataset (like ImageNet) to demonstrate the effectiveness and efficiency of the proposed DOS.

---- Acknowledgement ----
After reading the response, further clarification and discussion addressed most of the concerns in the following perspectives,
- Provided the specific definition of diversity and explained the relationship with clustering-based sampling; Complete the analysis on the outliers sampled by diversification target; Revised some unrigorous statements; Provided the large-scale experimental verification on the ImageNet dataset;

Overall, regarding the quality and the technical novelty of this work, the reviewer decided to raise the score accordingly.

**Questions:**

Please also refer to the weakness part for the general concerns. The following questions are more specific to the clarification and the reviewer hope these question can help the authors to improve the writing and presentation of this work.
1. As for the critical motivation ("However, the OOD samples selected solely based on uncertainty can be biased towards certain classes or
domains, which may fail to capture the full distribution of the auxiliary OOD dataset."), the intuitive illustration is based on a toy example in 2D space, it could be better to refer to some convincing results which also show the same problem with Figure 1c.
2. For the "imbalanced performance of OOD detection" pointed out in the same sentence, could the authors provide some empirical evidence?
3. The critical observation shows that "outlier subset comprising data from more clusters results in better OOD detection", but what is the relationship between the cluster with diversity?
4. Based on the previous question, could the authors provide a more detailed or specific conceptual definition of diversity? and clearly state how to measure the diversity in the motivation part to make that part more convincing.
5. It seems that the current version of the work does not provide the corresponding experimental part for supporting the claimed "efficient" of the proposed DOS, like the sampling efficiency if I understand it correctly.
6. In addition to using the CIFAR-100 as the ID dataset, it could be better to use the ImageNet dataset as ID dataset and use a large-scale ImageNet 21k as auxiliary outliers to verify the scalability of the proposed DOS.

---

> ### Author Response · Authors · 2023-11-19
> **Response to Reviewer HJAj (1/2)**
>
> Thank you for the constructive and elaborate feedback. Please find our response below:
>
> ## 1. Definition of diversity, mechanism of the clustering-based sampling [W2,Q4]
> Thank you for pointing out the ambiguous description. Here, we provide a rigorous definition of diversity and show how clustering-based sampling promotes diversity.
>
> Given a set of all OOD examples $\mathcal{D} = { \lbrace \mathbf{x}\_i \rbrace}^{N}\_{i=1}$, our goal is to select a subset of $K$ examples $\mathcal{S} \subseteq \mathcal{D}$ with $|\mathcal{S}| = K$. We define that the selected subset is diverse, if $\mathcal{S}$ satisfies the condition: $$\delta(\mathcal{S}) = \frac{1}{K}\sum\_{x\_{i} \in \mathcal{S}} \min _{j \neq i} d\left(x\_{i}, x\_{j}\right) \geq C,$$ where $d$ denotes a distance function in the space and $C$ is a constant that controls the degree of diversity. A larger value of $C$ reflects a stronger diversity of the subset. A greedy method to promote diversity is directly maximizing the left term $\delta(\mathcal{S})$ among all possible solutions. However, finding the optimal solution requires high computational complexity. And, it will be challenging to maintain the informativeness of selected examples, as you are concerned in W3.
>
> Therefore, we adopt the K-means clustering algorithm to keep the diversity of the selected subset. Intuitively, the clustering algorithm avoids repeatedly selecting examples in the same cluster, i.e., similar examples [1,2]. In particular, the distance between two examples from different clusters is lower bounded by the minimum distance between the boundary of the two clusters. By enlarging the separability of clusters (the goal of clustering), we can then increase the lower bound of $\delta(S)$, i.e., $\mathcal{C}$. In this way, we show the relationship between clustering and diversity.
>
> In addition, we adopt the Calinski-Harabasz (CH) index as a proxy to measure the diversity of the subset (see Figure 3). In particular, the CH index is the ratio of the between-cluster sum of squares (BCSS) and the within-cluster sum of squares (WCSS). A higher value of the CH index reflects a larger between-cluster distance and smaller within-cluster distance, leading to a higher value of $\mathcal{C}$.
>
> ## 2. Diversity may introduce easy outliers. [W3]
> Yes, promoting diversity in sampling would include some outliers that are easier than those selected by greedy sampling. As shown in Subsection 2.2, the hardest examples selected by the greedy sampling might be similar (See Figure 2a), leading to suboptimal performance in OOD detection (See Figure 2b). Therefore, we propose to sample **the hardest examples from different clusters**, which introduces diversity among the selected examples. Despite the fact that those diverse examples might not be the hardest globally, they are sufficiently informative to support the OOD detection task, as verified in Tables 1 and 2.
>
> ## 3. Convincing results for the bias of greedy sampling in Figure 1c [Q1]
> To support Figure 1c, we design an empirical analysis (see Subsection 2.2) to show the biased distribution caused by greedy sampling. As shown in Figure 2a, the greedy strategy leads to biased sampling, which exhibits an imbalanced distribution of outliers over the six clusters. To make it clearer, we add a note in the caption of Figure 1c.
>
> ## 4. Empirical evidence for the imbalanced OOD detection of biased sampling [Q2]
> Thank you for pointing out the ambiguous word. For clarification, we refer to "imbalanced" as "suboptimal" performance of OOD detection in the sentence. As shown in Figure 2b, the biased sampling leaves a large portion of OOD samples overconfident, which is suboptimal compared to the uniform sampling. In the revised version, we replace the "imbalanced" with "suboptimal" in Section 1.
>
> ## 5. The relationship between the cluster with diversity [Q3]
> We guess your concern is about how diverse clusters affect OOD detection. In Figure 2b, we provide an empirical analysis to show the advantage of uniform sampling, compared to biased sampling. The results show that the uniform strategy with max diversity achieves a much lower FPR95 than the biased strategy, which demonstrates the critical role of diversity in sampling.

---

> > ### Author Response · Authors · 2023-11-19
> > **Response to Reviewer HJAj (2/2)**
> >
> > ## 6. Efficiency of DOS [Q5]
> > Thank you for pointing out the potential for misunderstanding. The efficiency of DOS refers to the efficient training from fast convergence, instead of sampling efficiency. In particular, models trained with DOS converge to the optimal performance in much fewer epochs. As shown in Figure 5, our method achieves impressive performance at the 70th epoch, while the performances of random sampling and NTOM are not converged at the 100th epoch, with a significant gap to our method. More details can be found in Section 5 of the manuscript.
> >
> > ## 7. Evaluation on the large-scale benchmark [W4,Q6]
> > In addition to CIFAR100, we also conduct experiments on ImageNet-10 as the large-scale benchmark, following recent work on OOD detection [3]. The details of the experimental setting are introduced in Appendix C and the corresponding results are presented in Table 2. Our method achieves the best performance, surpassing the competitive OE by 3.07% in average FPR95. To avoid any confusion, we add a sentence in Subsection 4.1 to list all datasets we used in experiments.
> >
> > [1] Deep batch active learning by diverse, uncertain gradient lower bounds. ICLR'20.
> >
> > [2] Diverse mini-batch active learning. arXiv'19.
> >
> > [3] Delving into Out-of-Distribution Detection with Vision-Language Representations. NeurIPS'22.

---

> > ### Comment · Reviewer_HJAj · 2023-11-20
> > **Thanks for the responses**
> >
> > Dear Authors,
> >
> > Thanks for the responses with additional results and changes! I have read them and think most of the points are better clarified. As for the role of diversity, some related works [1,2] also discuss it with outlier exposure in OOD detection and implicitly/explicitly pursue it in the algorithm design, although not in a sampling-based way. It would be better if the authors could further discuss them, as they share similar conceptual pursuits. [1] Learning to Augment Distributions for Out-of-Distribution Detection. NeurIPS 2023. [2] Diversified Outlier Exposure for Out-of-Distribution Detection via Informative Extrapolation. NeurIPS 2023.
> >
> > Best regards,
> >
> > Reviewer HJAj

---

> > > ### Author Response · Authors · 2023-11-20
> > > **Discussion of related work**
> > >
> > > Thank you for the timely response and for contributing to the two related works. Here, we provide a detailed discussion to show the differences between our method and the two works.
> > >
> > > In particular, DAL [1] augments the auxiliary OOD distribution with a Wasserstein Ball. Their goal is to approximate the unseen OOD distribution via augmentation, while our method is designed to mitigate the biased distribution caused by the greedy sampling (and keep informative). DivOE [2] augments the informativeness of the auxiliary outliers via informative extrapolation, which generates new outliers closer to the decision boundary. Compared to our method, the diversity in their paper refers to generating more informative outliers, which are different from ours.
> > >
> > > In the revised version, we cite these two papers and add discussion in the related work.
> > >
> > >
> > > [1] Learning to Augment Distributions for Out-of-Distribution Detection. NeurIPS'23.
> > >
> > > [2] Diversified Outlier Exposure for Out-of-Distribution Detection via Informative Extrapolation. NeurIPS'23.

---

> > > > ### Comment · Reviewer_HJAj · 2023-11-21
> > > > **Thanks for the discussion**
> > > >
> > > > Dear Authors,
> > > >
> > > > Thanks for the responses with additional discussion! With further clarifications and the discussion on the related works, the previous concerns are well-addressed as follows:
> > > > - Provided the specific definition of diversity and explained the relationship with clustering-based sampling;
> > > > - Complete the analysis on the outliers sampled by diversification target;
> > > > - Revised some unrigorous statements;
> > > > - Provided the large-scale experimental verification on the ImageNet dataset;
> > > >
> > > > Overall, regarding the quality and the technical novelty of this work, the reviewer decided to raise the score accordingly after going through all the previous details again. Hope the previous comments can better improve the draft.
> > > >
> > > > Best regards,
> > > >
> > > > Reviewer HJAj

---

> > > > > ### Author Response · Authors · 2023-11-21
> > > > > **Many thanks!**
> > > > >
> > > > > Great thanks for your recognition and for raising the score. We are glad that our discussion finally addressed these concerns, which also improved the quality of this work.

---

### Official Review · Reviewer_bYrq · 2023-11-01

**Soundness:** 4 excellent
**Presentation:** 4 excellent
**Contribution:** 3 good
**Rating:** 8
**Confidence:** 4

**Summary:**

This paper proposes Diverse Outlier Sampling (DOS), a simple sampling approach for choosing diverse and informative outliers to use as an auxiliary OOD training dataset. The basic idea is simple: use k-means on normalized features to cluster the data and then per cluster identify the most informative outlier. Diversity is achieved since the k clusters collectively explain the full possible auxiliary OOD data but per cluster, we only choose the most informative outlier, which altogether ensures that the size of the auxiliary dataset can be controlled to be small (i.e., based on the number of clusters). Experimental results show that DOS works extremely well in practice.

**Strengths:**

- The paper is easy to follow.
- The basic idea of the proposed approach is very simple and elegant.
- The experimental results are compelling.

**Weaknesses:**

- By normalizing the feature vectors, if I understand correctly, Euclidean norm is used so that the normalized vector resides on the unit hypersphere. Is there any benefit to using specialized versions of k-means (and k-means related) algorithms for the hypersphere? For reference, there are versions of k-means and the Gaussian mixture model that are restricted to the hypersphere (technically, mixtures of von Mises-Fisher distributions). See, for instance, the papers by Banerjee et al (2005) and Kim (2021). More generally, some sort of sensitivity analysis with respect to using different clustering algorithms could be helpful.
- Figuring out how to set up experiments so that you could report error bars would be very helpful.

References:
- A Banerjee, I S Dhillon, J Ghosh, S Sra. Clustering on the Unit Hypersphere using von Mises-Fisher Distributions. JMLR 2005.
- M Kim. On PyTorch Implementation of Density Estimators for von Mises-Fisher and Its Mixture. arXiv 2021.

**Questions:**

See weaknesses.

---

> ### Author Response · Authors · 2023-11-19
> **Response to Reviewer bYrq**
>
> Thanks for your positive and valuable suggestions. Please find our response below:
> ## 1. Sensitivity analysis on clustering algorithms [W1]
> Thank you for the great suggestion. Here, we conduct sensitivity analysis to compare the performance of various clustering algorithms, including Agglomerative [1], K-Means++ seeding [2,3], Spherical K-Means, moVMF soft, and moVMF hard [4,5]. The experiments are conducted with the large benchmark (IN10-IN990). We keep the same training setting as in the manuscript. In addition to the OOD detection performance, we also present clustering time in each iteration, which is also a key factor of choosing clustering algorithm.
>
> As shown in the Table below, K-means (our choice) achieves the best performance among the compared algorithms. In addition, the two algorithms based on moVMF require too much time in each iteration, making it non-trivial to implement in this case. The resting algorithms cannot achieve comparable performance to ours, while they three obtains a little higher efficiency. We add the empirical results into Appendix D of the manuscript.
>
>
> |         Metric         | Agglomerative | K-Means++ seeding | Spherical K-Means | moVMF soft | moVMF hard | K-Means (Ours) |
> |:----------------------:|:-------------:|:-----------------:|:-----------------:|:----------:|:----------:|:--------------:|
> |   FPR95 $\downarrow$   |     38.39     |       48.19       |       24.11       |     -      |     -      |   **20.71**    |
> | Time Cost $\downarrow$ |   **0.05s**   |       2.0s        |       3.1s        |   242.1s   |   243.0s   |      4.5s      |
>
>
>
> ## 2. Presentation of error bars [W2]
> Thank you for pointing this out. We add the error bars in the revised version.
>
> [1] Ward’s Hierarchical Agglomerative Clustering Method: Which Algorithms Implement Ward’s Criterion? Journal of Classification 2014.
>
> [2] K-Means++: The Advantages of Careful Seeding. SODA'07.
>
> [3] Deep Anomaly Detection under Labeling Budget Constraints. ICML'23.
>
> [4] Clustering on the Unit Hypersphere using von Mises-Fisher Distributions. JMLR'05.
>
> [5] On PyTorch Implementation of Density Estimators for von Mises-Fisher and Its Mixture. arXiv'21.

---

> > ### Author Response · Authors · 2023-11-22
> > **Kind reminder to Reviewer bYrq**
> >
> > Dear Reviewer bYrq,
> >
> > Sorry to disturb you. We sincerely appreciate your valuable comments. Now we have followed your suggestions to conduct sensitivity analysis on various clustering algorithms (see Appendix D) and presented the error bars in Subsection 4.2. Please let us know if you still have any further concerns and we will be open to all possible discussions.

---

> > > ### Comment · Reviewer_bYrq · 2023-11-22
> > > **Response to author response**
> > >
> > > Thanks for taking the time to address my concerns with additional results. I have also looked over the other reviews and your responses to those reviews, and I have updated my score.

---

> > > > ### Author Response · Authors · 2023-11-22
> > > > **Many thanks!**
> > > >
> > > > Great thanks for your recognition and for raising the score. We are glad that our responses addressed these concerns, which also improved the quality of this work.

---

### Author Response · Authors · 2023-11-19
**General Response**

We thank all the reviewers for their time, insightful suggestions, and valuable comments. We are glad that **ALL** reviewers appreciate that our work is **simple** and **effective** with **compelling** results on **comprehensive** benchmarks. We are also encouraged that reviewers find this work focuses on an **important** and **practical** question (HJAj) with **well motivation** (BvLD), and the method is **elegant** (bYrq) and **novel** (HJAj). Besides, reviewers recognize that the writing is **clear** (bYrq,HJAj).

In the following responses, we have addressed the reviewers' comments and concerns point by point. The reviews allow us to strengthen our manuscript and the changes$^1$ are summarized below:
- Added sensitivity analysis on clustering algorithm in **Appendix D**. [bYrq,BvLD]
- Added definition of diversity in **Subsection 2.2** and **Appendix E**. [HJAj]
- Clarified convincing results for Figure 1c in its **caption**. [HJAj]
- Replaced the “imbalanced” with “suboptimal” in **Section 1**. [HJAj]
- Added description of experimental datasets in **Subsection 4.1**. [HJAj]
- Added two related works in **Appendix A**. [HJAj]
- Added error bars in **Subsection 4.2**. [bYrq]
- Added difference between diverse sampling of active learning and the proposed method in **Appendix A**. [BvLD]
- Fixed typo in **Equation 2**. [BvLD]

---
$^1$ For clarity, we highlight the revised part of the manuscript in **blue** color.

---

### Public Comment · ~Wenyu_Jiang1 · 2026-05-23
**Update First-Author Affiliation Order**

My university’s PhD thesis regulations require that any published work included in the dissertation must list my current institution as the primary affiliation.
To comply with this rule, I have changed the order of my affiliations from

1 Department of Statistics and Data Science, Southern University of Science and Technology

2 State Key Laboratory for Novel Software Technology, Nanjing University

to:

1 State Key Laboratory for Novel Software Technology, Nanjing University

2 Department of Statistics and Data Science, Southern University of Science and Technology

This change affects only the affiliation sequence. The author list, corresponding author designation, and all other details remain unchanged.

This change has obtained approval from the corresponding author, Prof. Hongxin Wei, who fully supports this modification.

---

### Meta-Review · Area_Chair_1odV · 2023-12-10

**Metareview:**

This paper proposes Diverse Outlier Sampling (DOS), a simple sampling approach for choosing diverse and informative outliers to use as an auxiliary OOD training dataset. The idea is simple: using K clusters to group OOD data and choose the most informative outliers. Experiments show that DOS works well in practice.

During the rebuttal, the authors have well addressed reviewers' concerns and all reviewers agreed to accept this work.

**Justification For Why Not Higher Score:**

This idea is simple, intuitive and effective. However, the general impact of this work is not transformative.

**Justification For Why Not Lower Score:**

All reviewers unanimously agree to accept this work.

---

### Decision · Program_Chairs · 2024-01-16

Accept (poster)